# Differences in referral path, clinical and radiographic outcomes between seronegative and seropositive rheumatoid arthritis Mexican Mestizo patients: A cohort study

**Guillermo Arturo Guaracha-Basáñez**[ID][☯], **Irazú Contreras-Yáñez**[☯], **Ana Belén Ortiz-Haro**[☯], **Virginia Pascual-Ramos**[ID][☯]*

Department of Immunology and Rheumatology, Instituto Nacional de Ciencias Médicas y Nutrición Salvador-Zubirán (INCMyN-SZ), Mexico City, Mexico

☯ These authors contributed equally to this work.

* virtichu@gmail.com

## Abstract

### Background

The study compared the referral path, the first two-year clinical outcomes, and the first five-year radiographic outcomes between seronegative patients (SNPs) from a recent-onset rheumatoid arthritis dynamic cohort initiated in 2004 and seropositive patients (SPPs). Predictors of incidental erosive disease were investigated.

### Patients and methods

Up to March 2023, one independent observer reviewed the charts from 188 patients with at least two years of clinical assessments and up to five years of annual radiographic assessments. SNPs were defined when baseline RF and ACPA serum titers were within local normal ranges. The erosive disease was defined on hand and/or foot radiographs when at least one unequivocal cortical bone defect was detected. The incidental erosive disease was defined in baseline erosive disease-free patients who developed erosions at follow-ups. Multivariate Cox regression analyses identified hazard ratios (95% confidence interval) for factors to predict incidental erosive disease.

### Results

There were 17 (9%) SNPs, and they had a shorter time from symptoms onset to first physician evaluation, visited a lower number of physicians, and received less intensive treatment at referral and during the first years of follow-up than SPPs. Also, they had fewer 0–66 swollen joints and were less frequently persistent on therapy. The erosive disease was detected only in SPPs, and its frequency increased from 10.1% at baseline to 36.1% at the five-year radiographic assessment. There were 53 (31.4%) patients with incidental erosive disease, and differences between SPPs and SNPs were statistically significant at the feet location.

**Data Availability Statement:** The data that support the findings of this study are not openly available due to reasons of sensitivity and privacy. However, our complete data are available from the corresponding author upon reasonable request. Data requests can also be placed with the chair of the Research Ethics Committee. Dr. Sergio C. Hernández Jiménez is currently the chair of the Research Ethics Committee (e-mail: sergio. hernandezj@incmnsz.mx). The details that ensure long-term data storage and availability include (but are not limited to): duplicate charts (paper and electronic), database updated on regular bases, regular electronic data backup and staff backup for the person in charge of the data availability.

**Funding:** The author(s) received no specific funding for this work.

**Competing interests:** The authors have declared that no competing interests exist.

Incidental erosive disease was predicted by baseline ACPA, ESR, substantial morning stiffness, and cumulative CRP.

## Conclusions

SNPs showed mild differences in their referral path and clinical outcomes compared to SPPs. However, erosive disease was detected only in SPPs and was predicted by baseline and cumulative clinical and serologic variables.

## Introduction

Rheumatoid arthritis (RA) is increasingly recognized as a highly heterogeneous syndrome whose expression encompasses various clinical phenotypes, responses to treatment, and outcomes [1, 2]. The recognition of RA-associated autoantibodies, with the rheumatoid factor (RF) and antibodies against citrullinated proteins (ACPA) being the most acknowledged ones, supports the auto-immune nature of the disease [3]. RA patients who are positive for serum RF and/or ACPA are defined as "seropositive" patients (SPPs) and are considered to display a different etiology, disease nature, and course compared to the so-called "seronegative" patients (SNPs) who had been described as a not well-characterized group [3].

A recent review of seronegative RA highlights a high variability in the frequency of seronegative disease within general RA cohorts [1, 4]. At the same time, an increased incidence is observed due to changes in demographics and environmental risk factors [1, 5, 6]. Also, the review addresses relevant differences in the susceptibility loci [1], risk factors [1, 5–8], pathogenic pathways [1], and the benefits attributed to specific drug combinations [9, 10] between the two subsets of RA. In addition, it emphasizes a shorter and more abrupt pre-clinical history of SNPs compared to that of SPPs [1, 11, 12], a precocious and common involvement of tendons [1, 13], and better clinical, serological, and radiographic outcomes [1, 14, 15], but persistent pain [1, 16]. However, the current knowledge regarding the peculiarities of seronegative RA has been hampered by diagnostic difficulties, treatment differences in patients' management, and conceived based on clinical evidence obtained from primarily Caucasian populations, which limits the comprehensiveness of the topic. It is worth mentioning that the seropositive subgroup highly represents RA patients. In this subgroup, distinctive characteristics had been described among RA patients from the Latin-American region (LATAM) compared to Caucasian patients. Some affect socio-demographics, such as younger age at presentation (almost ten years earlier) and an extreme female preponderance [17]. Others extend to patients' perceptions and views of particular outcomes primarily influenced by nationality, ethnicity, and cultural background [18]. In addition, patients from non-developed countries experienced substantial delays in diagnosis, which is usually observed as several months but might extend to years in Latin American countries, and results in worse outcomes and patterns of radiographic progression, which begin early and persist after a prolonged follow-up [19, 20]. The differential disease expressions observed among (primarily) seropositive RA patients from LATAM might extend to patients with negative autoantibodies and justify the current study.

In 2004, we began assembling a recent-onset dynamic cohort of RA patients at a national referral center for rheumatic diseases in Mexico City, the Instituto Nacional de Ciencias Médicas y Nutrición Salvador-Zubirán (INMCyN-SZ). Patients included in the cohort had

complete periodic assessments and were considered appropriate candidates for achieving the following study objectives:

1. - To compare the referral path, baseline characteristics, and the first two years of follow-up clinical outcomes between SNPs from a recent-onset RA dynamic cohort and SPPs.

2. - To compare the two groups' baseline and annual radiographic outcomes during the first five years of follow-up.

3. - To identify predictors of incidental erosive disease, with a particular interest in the baseline presence of RF and ACPA.

## Materials and methods

### Setting and study population: The recent-onset dynamic RA cohort

Patients whose data were analyzed were identified from the recent-onset dynamic RA cohort initiated in 2004 at the INCMyN-SZ.

Patients who had ≤ 12 months of symptom duration, at least one swollen joint, and no other rheumatic diagnosis but RA entered the recent-onset cohort [21]. At the baseline evaluation, the primary rheumatologist recorded the patient's real-world referral path/journey to the cohort and a complete medical history, including demographic data, RF, and ACPA. At baseline and follow-ups, all the patients had standardized rheumatic assessments (66/68 swollen/tender joint counts, DAS28-ESR [22], and physician-overall disease visual analog scale [VAS]), patient-reported outcomes measures (PROMs) recorded (disability as per the Health Assessment Questionnaire disability index [HAQ-DI], quality of life (QoL) as per the Short-Form 36 [SF-36], fatigue [as per SF-36], pain [as per a pain-VAS], and patient overall disease status [overall-VAS]) [23], assessment of comorbidities [24], medication behavior [21] and treatment; in addition, laboratory parameters were obtained.

All the patients in the cohort were evaluated at baseline and every two months during the first two years of follow-up. After that, visits were scheduled every two, four, or six months (mandatory for all the patients), depending on the patients and disease characteristics. The rheumatologist in charge of patients' follow-ups prescribed treatment. It was Treat-to-Target (T2T) oriented and primarily with conventional disease-modifying anti-rheumatic drugs (DMARDs) with/without glucocorticoids.

Finally, radiographs of hands and feet were scheduled at baseline and annually after that.

### Study design and data collection

The current study has a cohort design. Through March 2023, the cohort comprised 237 RA patients. One hundred eighty-eight patients (79.3%) had at least two years of follow-up, the minimum time required for a significant clinical patient´ follow-up. Also, 188 (79.3%), 187 (78.9%), 157 (66.2%), 161 (67.9%), and 144 (60.8%) patients had consecutive annual radiographs and their data will be used to compare erosive disease prevalence and annual incidences during the first five years of follow-up between RA subsets (objective two) and to examine predictors of incidental erosive disease (objective three).

Up to the cut-off, 10 (5.3%) patients were deceased, 43 (22.9%) were lost to follow-up, and 135 (71.8%) were currently active in the cohort (Fig 1). Interestingly, there were differences between seronegative and seropositive groups in the number (%) of patients currently active (11 [45.8%] vs. 124 [75.6%]), lost to follow-up (13 [54.2%] vs. 30 [18.3%]) and dead (0 vs. 10 [6.1%]) (p≤0.0001).

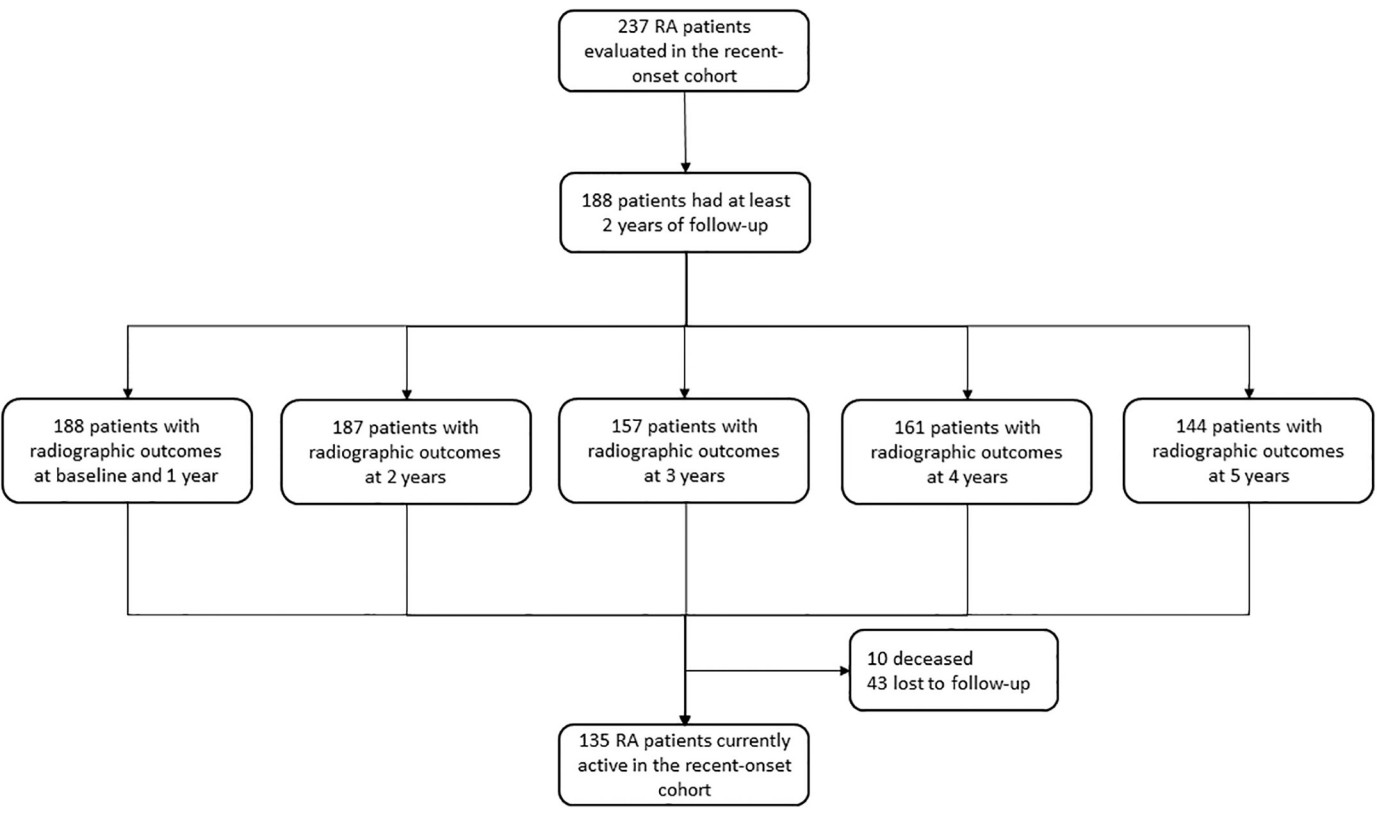

**Fig 1. Study flow-chart.**

Ending in May 2023 and up to the last follow-up relevant to the study objectives or death, all the charts were retrospectively reviewed by a single trained data abstractor who corroborated the integrity of the data collected.

## Definitions

A senior X-ray technician was in charge of performing digitized images of radiographs of the hands (posterior-anterior and oblique views) and feet (posterior-anterior view) (Digitized, CR900 Direct view, Serie number:3172897, 120V, 10A, Kodak). Hand and foot radiographs were read chronologically by a radiologist and a rheumatologist, who were neither blinded to the patient's clinical context nor to each other evaluation. *RA was classified as erosive* if both physicians identified at least one unequivocal cortical bone defect or break after carefully reviewing radiographs, and no formal validated scoring system was used. Disagreement was resolved by consensus. For the current report, the erosive disease was evaluated at baseline and then annually for up to five years of follow-up. *Incidental erosive disease* was defined in baseline erosive disease-free patients who developed erosions at follow-ups.

RF levels were determined by nephelometry and ACPA by second-generation (from 2004 to 2012) and third-generation (from 2013 onwards) ELISA. At the baseline evaluation, SNPs were defined when serum RF and ACPA titers were below local reference values. At the baseline evaluation, SPPs were defined when serum RF and/or ACPA titers were above local reference values.

## Sample size estimation

Data from all the patients from the cohort up to March 2023, with recent-onset RA and at least two-year follow-up information, were analyzed.

We performed a post hoc power analysis to achieve the study objectives. We retrospectively estimated the power achieved considering the probability of error α equal to 0.05 and the results obtained: a difference in cumulative swollen joints and radiographic outcomes between SNPs and SPPs and the hazard ratio (HR) for ACPA, substantial morning stiffness and cumulative CRP to predict incidental erosive disease. In all cases, the power achieved with the 188 patients whose data were analyzed was ≥ 0.80. However, the power achieved for baseline ESR HR to predict incidental erosive disease was 0.57.

We used G power and Epidat software for post-hoc power analysis.

## Statistical analysis

Descriptive statistics were used with frequencies and percentages for dichotomous variables and median (percentile [p] 25–75) for continuous variables (non-normal distribution).

The patients' characteristics at referral to the cohort, at the baseline evaluation, and during follow-ups were compared between SNPs and SPPs and between patients with incidental erosive disease and patients erosive disease-free at the last follow-up, using appropriate tests: the $X^2$ test for the categorical variables and the Mann-Whitney U test for continuous variables (non-normal distribution).

Cumulative clinical outcomes were restricted for up to the first five years of follow-ups. The following outcomes were considered: disease activity (DAS28-ESR), pain (pain-VAS), patient-overall disease-VAS, fatigue (SF-36 corresponding component), disability (HAQ-DI), QoL (SF-36), treatment (glucocorticoids use and the number of DMARDs/patient), persistence on therapy and Charlson score [24]. Cumulative clinical outcomes behavior (for continuous variables) was summarized as the Area Under the Curve (AUC), calculated by the trapezoid method and presented standardized by the length of the study period evaluated [25]. Dichotomous variables (prednisone use [or equivalent], persistence on therapy, and morning stiffness duration greater than 15 min) were summarized if present during cumulative follow-ups.

Multivariate Cox regression analysis estimated HRs (95% confidence interval [CI]) to define variables predictive of incidental erosive disease. We conceived models that considered variables at cohort entry (Model 1), cumulative variables (Model 2), and a combination of baseline and cumulative variables (Model 3). Variables' inclusion was based on their statistical significance in the univariate analysis and clinical relevance. Considering that more than fifty variables were tested, the probability of obtaining spurious associations was higher than the nominal alpha value of 0.05 for each test. Bonferroni adjustment was considered to protect against this risk (p≤0.0009). The variables included in the model were considered simultaneous independent variables after revising collinearity (one variable was excluded if rho≥0.70). Models were repeated after RF and ACPA were forced. A test-based backward selection procedure defined the variables significantly associated with incidental erosive disease (dependent variable). The Log Likelihood function is reported as a measure of the model goodness of fit.

Missing data varied from 0% (socio-demographics, baseline characteristics, and cumulative follow-ups) to 13% (real-life referral journey), and no imputation was performed.

All statistical tests were two-sided and evaluated at the 0.05 significance level. All analyses were performed using SPSS (version 21.0, IBM Corp., Armonk, NY, USA).

## Ethics

The Institutional Review Board approved the study (Comités de Ética e Investigación del INCMyN-SZ: IRE-274-10/11-1). When entering the clinic, all the patients provided written informed consent for clinical follow-ups. They also offered additional written permission to review each patient´s chart and present their data in scientific publications and forums.

## Results

### Population characteristics (Tables 1 and 2)

Overall, patients were primarily middle-aged females (167 [88.8%]), with (median, p25-75) 12 (9–15) years of formal education and a medium-low socioeconomic level (168 [89.4%]). Patients mentioned a short time from symptoms onset to first physician evaluation (18.5 [2–59.8] days), performed in the minority by a specialist (36 [22.2%]). Half the patients were indicated DMARDs by the first physician who evaluated them, and 58 (30.9%) glucocorticoids. Among the 94 patients on DMARDs, 64 (68%) were receiving methotrexate, 14 (15%) chloroquine, and two each (2.1%) leflunomide and sulfasalazine. In contrast, only 12 (12.8%) patients were on combined DMARDs.

At the baseline evaluation in the recent-onset cohort, patients had a short disease duration (5 months [3–6.8]) and substantial clinical and serological disease activity, translating into a significant impact on the PROMs. In addition, 187 patients (99.5%) received combined DMARDs and 98 (52.1%) glucocorticoids (Table 1). The most frequent DMARD combinations were methotrexate and chloroquine in 148 (78.8%) patients, methotrexate, chloroquine and sulfasalazine in 20 (10.6%), methotrexate and leflunomide in 4 (2.1%) and methotrexate and sulfasalazine in 2 (1.1%) patients (data available for 174 patients).

Finally, the number of American College of Rheumatology (ACR) classification criteria for RA was 5 (5–6) [26].

Table 2 summarizes cumulative outcomes during the first two years of follow-up and highlights a significant improvement in joint counts, acute reactant phase determinations, and PROMs, translating into mild disease activity, the absence of disability, pain under control, and good QoL. Meanwhile, all the patients received increased combined DMARDs (2.6 [2–3]) and 114 (60.6%) glucocorticoids. The most frequent DMARD combinations included two DMARDs prescribed in 110 (63.2%) patients, among whom the most frequent combination was methotrexate and chloroquine in 60 patients. Also, 60 (35.4%) additional patients received three DMARDs combined (35 received methotrexate, chloroquine, and sulfasalazine). Finally, four (2.2%) patients received four DMARDs combined (methotrexate, chloroquine, sulfasalazine, and leflunomide) (data available for 174 patients).

Finally, 114 (60.6%) patients were persistent in therapy during the period evaluated.

### Comparison of the sociodemographic characteristics, referral path, baseline disease-related characteristics, and first two years of follow-up clinical outcomes between SNPs and SPPs (Tables 1 and 2)

There were 17 patients (9%) classified as seronegative and 171 (91%) as seropositive. Baseline socio-demographic characteristics were similar between groups. Meanwhile, the time from symptoms onset to the first physician evaluation (days) was longer in SPPs (20 [5–60] vs. 11.5 [1.8–33.5], p = 0.015), who also visited a higher number of physicians before the first evaluation at the cohort (1 [0–2] vs. 0 [0–1], p = 0.001), had more frequently glucocorticoids at cohort referral (95 [55.6%] vs. 3 [17.6%], p = 0.004) and a higher number of DMARDs/patient

**Table 1. Sociodemographic characteristics, patient´s referral path, baseline RA-related outcomes in the overall population, and their comparison between SNPs and SPPs.**

| | Overall population (N = 188) | SNPs (N = 17) | SPPs (N = 171) | p |
|---|---|---|---|---|
| **Socio-demographic characteristics** | | | | |
| Years of age | 38 (27–49) | 41 (25–58) | 38 (27–48.4) | 0.531 |
| Female[1] | 167 (88.8) | 16 (94.1) | 151 (88.3) | 0.698 |
| Years of formal education | 12 (9–15) | 12 (9.5–15) | 12 (9–15) | 0.511 |
| Patients with current or past smoking[1] | 18 (9.6) | 1 (5.9) | 17 (9.9) | 1 |
| Patients married or living together[1] | 89 (47.3) | 11 (64.7) | 78 (45.6) | 0.202 |
| Medium-low socioeconomic level[1] | 168 (89.4) | 14 (82.4) | 154 (90.1) | 0.399 |
| **Real-life referral path/journey*** | | | | |
| Time from symptom onset to first physician evaluation (days) | 18.5 (2–59.8) | 11.5 (1.8–33.5) | 20 (5–60) | **0.015** |
| Patients first evaluated by a specialist (vs. primary care physician)[1] | 36 (22.2) | 4 (40) | 31 (21.1) | 0.231 |
| Physicians visited before the first evaluation at the cohort | 1 (1–2) | 0 (0–1) | 1 (0–2) | **0.001** |
| Patients on glucocorticoids at referral to the cohort[1] | 58 (30.9) | 2 (11.8) | 56 (32.7) | 0. 099 |
| Time on glucocorticoids at referral to the cohort[2] (days) | 32.1 (14.5–85.4) | 30 (17.1–30) | 32.1 (14.5–86.5) | 0.798 |
| Patients on DMARDs at referral to the cohort[1] | 94 (50) | 6 (35.3) | 88 (51.5) | 0.309 |
| Time on DMARDs at referral to the cohort[2] (days) | 22.5 (6.7–61.2) | 15.9 (0.7–102) | 22.5 (7.2–61.6) | 0.653 |
| Number of DMARDs/patient[2] | 1 (1–1) | 1 (1–1) | 1 (1–1) | 0.831 |
| **Baseline RA-related characteristics** | | | | |
| BMI | 25.9 (22.9–28.8) | 24.7 (22.3–28.4) | 25.9 (22.8–28.8) | 0.356 |
| Disease duration (months) | 5 (3–6.8) | 3.3 (2.4–7.4) | 5.1 (3.4–6.8) | 0.156 |
| Patients with erosive disease[1] | 19 (10.1) | 0 | 19 (11.1) | 0.226 |
| Swollen joints (0–66) | 16 (10–24) | 14 (6–17.5) | 16 (10–25) | 0.065 |
| Tender joints (0–68) | 18 (11–28) | 14 (10–23.5) | 18 (11–28) | 0.361 |
| ESR mm/H | 21 (10–39) | 18 (11–22.8) | 22.5 (10–40.8) | 0.108 |
| CRP, mg/dL | 0.6 (0.2–2.5) | 0.3 (0.1–1.4) | 0.7 (0.3–2.5) | 0.125 |
| Patient-overall-disease VAS | 52 (29.3–74.8) | 36 (15–66.7) | 53 (31–75) | 0.093 |
| DAS28 | 5.7 (4.5–6.8) | 5.4 (4–6) | 5.8 (4.6–6.9) | 0.134 |
| Physician-VAS | 35 (25–48) | 34 (15–43.5) | 35 (25–49) | 0.241 |
| HAQ-DI score | 1.3 (0.8–2) | 1 (0.5–1.4) | 1.4 (0.8–2.1) | 0.065 |
| SF-36 score, physical component | 37 (24–55) | 39 (27.5–61) | 36 (23–55) | 0.157 |
| SF-36 score, mental component | 45 (30–61) | 63 (35.5–72) | 45 (29–59) | 0.078 |
| Pain-VAS | 50 (28–73) | 43 (21.5–62.5) | 50 (31–74) | 0.072 |
| Fatigue score (0–100) (lower scores = more fatigue) | 50 (35–60) | 50 (39–67.6) | 50 (35–60) | 0.193 |
| Patients with substantial morning stiffness[1] (over 15 min) | 154 (81.9) | 14 (82.4) | 140 (81.9) | 1 |
| Charlson score | 1 (1–1) | 1 (1–1) | 1 (1–1) | 0.457 |
| Patients indicated glucocorticoids[1] | 98 (52.1) | 3 (17.6) | 95 (55.6) | **0.004** |
| Patients indicated DMARDs[1] | 187 (99.5) | 17 (100) | 170 (99.4) | 1 |
| Number of DMARDs/patient[2] | 2 (1–2) | 1 (1–2) | 2 (2–2) | **≤0.0001** |
| Number of ACR 1987 classification criteria | 5 (5–6) | 4 (4–5) | 5 (5–6) | **≤0.0001** |

Data presented as median (p25-75) unless

[1] = N° (%) of patients.

[2]Among those with the characteristic.

*Twenty-six missing data.

 

**Table 2. First two-year cumulative outcomes and their comparison between SNPs and SPPs.**

| | Overall population (N = 188) | SNPs (N = 17) | SPPs (N = 171) | p |
|---|---|---|---|---|
| Swollen joints (0–28) | 3.6 (2.2–5.2) | 2.8 (1.5–4.6) | 3.6 (2.2–5.4) | 0.100 |
| Swollen joints (0–66) | 4.4 (2.7–6.6) | 3 (1.5–4.8) | 4.6 (3–6.8) | **0.012** |
| Tender joints (0–28) | 3.2 (2–4.8) | 2.8 (1.8–4.5) | 3.2 (2–4.8) | 0.582 |
| Tender joints (0–68) | 4.6 (3–6.8) | 4 (2.4–5.9) | 4.8 (3–6.8) | 0.247 |
| ESR mm/H | 12.2 (8–20) | 10.2 (7.2–17.8) | 12.6 (8–20) | 0.307 |
| CRP, mg/dL | 0.5 (0.2–1.0) | 0.3 (0.1–0.7) | 0.5 (0.2–1.1) | 0.250 |
| Patient-overall-disease VAS | 14.7 (9.2–20.3) | 11 (5.2–19.4) | 15 (9.5–20.4) | 0.226 |
| DAS28 | 2.7 (2.1–3.3) | 2.5 (2.1–3) | 2.7 (2.1–3.3) | 0.309 |
| Physician-VAS | 10.4 (7–15.6) | 8.6 (6.2–13.4) | 10.4 (7.4–17.2) | 0.105 |
| HAQ-DI score | 0.3 (0.2–0.5) | 0.3 (0.2–0.4) | 0.3 (0.2–0.5) | 0.526 |
| SF-36 score, physical component | 74 (67–79.4) | 74.1 (67.6–83.6) | 73.8 (66.9–78.9) | 0.423 |
| SF-36 score, mental component | 77.8 (71.1–84.9) | 79.3 (71.3–89.3) | 77.6 (70.9–84.4) | 0.356 |
| Pain-VAS | 14.4 (8.6–19.1) | 15.2 (7.6–17.5) | 14.4 (9.1–19.3) | 0.467 |
| Fatigue score (0–100) | 67.6 (57.7–75.8) | 69 (56.8–79.5) | 67 (57.6–75) | 0.645 |
| Patients with substantial morning stiffness[1] | 6 (3.2) | 1 (5.9) | 5 (2.9) | 0.440 |
| Charlson score | 1 (1–1) | 1 (1–1) | 1 (1–1) | 0.967 |
| Patients on glucocorticoids[1] | 114 (60.6) | 5 (29.4) | 109 (63.7) | **0.008** |
| Patients on DMARDs[1] | 188 (100) | 17 (100) | 171 (100) | 1 |
| Number of DMARDs/patient[2] | 2.6 (2–3) | 2.2 (1.4–2.5) | 2.6 (2–3) | **0.001** |
| Patients persistent in therapy[1] | 114 (60.6) | 6 (35.3) | 108 (63.2) | **0.036** |

Data presented as median (p25-75) unless

[1] = N° (%) of patients.

[2] Among those with the characteristic.

*33 missing data.

(2 [2–2] vs. 1 [1–2], p≤0.0001), compared to their counterparts. Baseline clinical outcomes, acute reactant-phase determinations, and PROMs were similar between both groups, although some tendencies were observed toward more deteriorated outcomes among SPPs. Also, as expected, SNPs differed from SPPs in the number of ACR 1987 classification criteria for RA: 4 (4–5) vs. 5 (5–6), p≤0.001 (Table 1).

During the first two years of follow-up, SPPs showed a higher number of (0–66) swollen joints (4.6 [3–6.8] vs. 3 [1.5–4.8], p = 0.012) and received a higher number of DMARDs/patient (2.6 [2–3] vs. 2.2 [1.4–2.5], p≤0.001) compared to SNPs. Also, more SPPs received glucocorticoids (109 [63.7%] vs. 5 (29.6%), p = 0.008) and were persistent on therapy (105 [63.2%] vs. 6 [35.3%], p = 0.036), as summarized in Table 2.

### Radiographic outcomes (Table 3)

At baseline and follow-ups, erosive disease was detected only in SPPs. Table 3 summarizes the number (%) of patients with erosions in the overall population and compares baseline and annual incident radiographic outcomes between SNPs and SPPs. Overall, the number (%) of patients with erosive disease increased over follow-ups, from 10.1% at baseline to 36.1% at five-year radiographic assessment. At any time, erosions were detected more frequently in the feet. Differences in incidental erosive disease between SNPs and SPPs were statistically significant at the feet location.

 

**Table 3. Radiographic outcomes in the overall population and comparison between SNPs and SPPs.**

| Baseline radiographic outcomes | | | |
|---|---|---|---|
| | Overall population N = 188 | SNPs N = 17 | SPPs N = 171 | p |
| Patients with erosions | 19 (10.1) | 0 | 19 (11.1) | 0.226 |
| Patients with hand erosions | 5 (2.7) | 0 | 5 (2.9) | 1 |
| Patients with foot erosions | 16 (8.5) | 0 | 16 (9.4) | 0.369 |
| One year´s incidental radiographic outcomes | | | |
| | Overall population N = 188 | SNPs N = 17 | SPPs N = 171 | p |
| Patients with erosions | 35 (18.6) | 0 | 35 (20.5) | **0.046** |
| Patients with hand erosions | 15 (8) | 0 | 15 (8.8) | 0.368 |
| Patients with foot erosions | 28 (14.9) | 0 | 28 (16.4) | 0.081 |
| Two year´s incidental radiographic outcomes | | | |
| | Overall population N = 187 | SNPs N = 17 | SPPs N = 170 | p |
| Patients with erosions | 45 (24.1) | 0 | 45 (26.5) | **0.014** |
| Patients with hand erosions | 17 (9.1) | 0 | 17 (10) | 0.373 |
| Patients with foot erosions | 36 (19.3) | 0 | 36 (21.2) | **0.047** |
| Three year´s incidental radiographic outcomes | | | |
| | Overall population N = 157 | SNPs N = 13 | SPPs N = 144 | p |
| Patients with erosions | 46 (29.3) | 0 | 46 (31.9) | **0.011** |
| Patients with hand erosions | 23 (14.6) | 0 | 23 (16) | 0.218 |
| Patients with foot erosions | 37 (23.6) | 0 | 37 (25.7) | **0.039** |
| Four year´s incidental radiographic outcomes | | | |
| | Overall population N = 161 | SNPs N = 12 | SPPs N = 149 | p |
| Patients with erosions | 51 (31.7) | 0 | 51 (34.2) | **0.010** |
| Patients with hand erosions | 23 (14.5) | 0 | 23 (15.6) | 0.217 |
| Patients with foot erosions | 45 (28.3) | 0 | 45 (30.6) | **0.020** |
| Five year´s incidental radiographic outcomes | | | |
| | Overall population N = 144 | SNPs N = 10 | SPPs N = 134 | p |
| Patients with erosions | 52 (36.1) | 0 | 52 (38.8) | **0.014** |
| Patients with hand erosions | 19 (13.2) | 0 | 19 (14.2) | 0.359 |
| Patients with foot erosions | 46 (31.9) | 0 | 46 (34.3) | **0.031** |

Data presented as number (% of patients).

## Predictors of incidental erosive disease: RF and ACPA attribution

Fifty-three (31.4%) patients developed incidental erosive disease; their data were compared with those of 116 patients who were erosive disease-free and are summarized in Table 4. Briefly, socio-demographics and real-life referral journey characteristics were similar. However, patients with incidental erosive disease had more tender and swollen joint counts, higher acute reactant phase determination and DAS28, scored worse on the physician-VAS and PROMs (but pain and fatigue), had more frequently substantial morning stiffness and met a higher number of ACR 1987 classification criteria, all at the baseline evaluation. Cumulative outcomes confirmed greater disease activity (tender and joint counts, acute reactant phase determinations levels, DAS28, and physician-VAS), more deteriorated PROMs (but SF36, fatigue, and substantial morning stiffness), and more intensive treatment among the patients with incidental erosive disease.

Table 5 summarizes the multivariate Cox regression analysis results to define incidental erosive disease predictors. The table presents the variables included in each model, significant

**Table 4. Comparison of baseline characteristics, referral path, baseline clinical outcomes, and cumulative outcomes between patients with incidental erosive disease and patients erosive disease-free.**

| | Patients with incidental erosive disease, N = 53 | Patients incidental erosive disease-free, N = 116 | p |
|---|---|---|---|
| **Socio-demographic characteristics** | | | |
| Years of age | 36.6 (26.5–47.7) | 37.5 (26.7–48) | 0.914 |
| Female[1] | 48 (90.6) | 103 (88.8) | 1 |
| Years of formal education | 12 (9–14) | 12 (9–16) | 0.591 |
| Patients with current or past smoking[1] | 3 (5.7) | 11 (9.5) | 0.552 |
| Patients married or living together[1] | 27 (50.9) | 52 (44.8) | 0.508 |
| Medium-low socioeconomic level[1] | 48 (90.6) | 102 (87.9) | 0.794 |
| **Real-life referral path*** | | | |
| Time from symptom onset to first physician evaluation (days) | 15 (0–60) | 19.5 (2–58.8) | 0.705 |
| Patients first evaluated by a specialist (vs. primary care physician)[1] | 9 (20.5) | 24 (24) | 0.830 |
| Physicians visited before the first evaluation at the cohort | 1 (0–2) | 1 (0–1) | 0.100 |
| Patients on glucocorticoids at referral to the cohort[1] | 17 (32.1) | 33 (28.4) | 0.717 |
| Time on glucocorticoids at referral to the cohort[2] (days) | 36 (25–84.5) | 26 (13.5–86) | 0.424 |
| Patients on DMARDs at referral to the cohort[1] | 25 (47.2) | 58 (50) | 0.743 |
| Time on DMARDs at referral to the cohort[2] (days) | 30 (4–84) | 18.5 (3.7–58.4) | 0.571 |
| Number of DMARDs/patient[2] | 1 (0–2) | 1 (0–1) | 0.540 |
| **Baseline RA-related characteristics** | | | |
| BMI | 25.9 (23.7–29.2) | 25.2 (22.7–28.7) | 0.397 |
| Disease duration (months) | 5.4 (3.2–6.9) | 4.8 (3–6.8) | 0.334 |
| Swollen joints (0–66) | 17 (12–29) | 14 (9–21) | **0.004** |
| Tender joints (0–68) | 22 (15–29) | 15 (9–21) | **0.006** |
| ESR mm/H | 30 (19.5–57.5) | 18.5 (7–27.8) | **≤0.0001** |
| CRP, mg/dL | 2 (0.5–3.9) | 0.4 (0.2–1.5) | **≤0.0001** |
| Patients with seronegative status[1] | 0 | 17 (14.7) | **0.002** |
| Patient-overall-disease VAS | 56 (45.5–75.5) | 45 (27.3–74.8) | **0.035** |
| DAS28 | 6.3 (5.5–7) | 5.3 (4.1–6.4) | **≤0.0001** |
| Physician-VAS | 45 (30.5–55.5) | 31 (27–43) | **0.001** |
| HAQ-DI score | 1.6 (1.1–2) | 1.1 (0.6–1.8) | **0.003** |
| SF-36 score, physical component | 31 (21–45.5) | 42.5 (24.3–61) | **0.003** |
| SF-36 score, mental component | 38 (27.5–52) | 50 (30–65) | **0.003** |
| Pain-VAS | 54 (35–75.5) | 47 (24–70) | 0.062 |
| Fatigue score (0–100) (lower scores = more fatigue) | 45 (36.5–55) | 50 (35–60) | 0.708 |
| Patients with substantial morning stiffness[1] (above 15 min) | 52 (98.1) | 86 (74.1) | **≤0.0001** |
| Charlson score | 1 (1–1) | 1 (1–1) | 0.772 |
| Patients indicated glucocorticoids[1] | 29 (54.7) | 56 (48.3) | 0.508 |
| Patients indicated DMARDs[1] | 53 (100) | 115 (99) | 1 |
| Number of DMARDs/patient[2] | 2 (1.5–3) | 2 (1–2) | 0.067 |
| Number of ACR 1987 classification criteria for RA | 6 (5–6) | 5 (4–6) | **≤0.0001** |
| Patient´s follow-up (months) | 60 (60–60) | 60 (24–60) | 0.270 |
| **Cumulative outcomes** | | | |
| Swollen joints (0–28) | 4.4 (3.1–6.5) | 3 (1.6–4.6) | **≤0.0001** |
| Swollen joints (0–66) | 5.5 (3.3–7.9) | 2.8 (1.8–4.7) | **≤0.0001** |
| Tender joints (0–28) | 4 (2.6–5.7) | 2.6 (1.6–4.2) | **0.001** |
| Tender joints (0–68) | 4.8 (3.3–6.8) | 3.2 (1.8–4.7) | **≤0.0001** |

(*Continued*)

**Table 4.** (Continued)

| | Patients with incidental erosive disease, N = 53 | Patients incidental erosive disease-free, N = 116 | p |
|---|---|---|---|
| ESR mm/H | 17.3 (9.3–23.4) | 10 (5.7–15.2) | ≤0.0001 |
| CRP, mg/dL | 0.9 (0.5–1.7) | 0.4 (0.2–0.7) | ≤0.0001 |
| Patient-overall-disease VAS | 13 (9.1–18.3) | 10.3 (5.4–15.3) | 0.005 |
| DAS28 | 2.9 (2.4–3.4) | 2.2 (1.7–2.7) | ≤0.0001 |
| Physician-VAS | 12. 8 (8.6–18.6) | 7.5 (5.2–11.5) | ≤0.0001 |
| HAQ-DI score | 0.3 (0.2–0.5) | 0.2 (0.1–0.4) | 0.001 |
| SF-36 score, physical component | 72.5 (63.4–79) | 73.5 (60–81.2) | 0.603 |
| SF-36 score, mental component | 77.2 (67.8–84) | 77.9 (65.2–85.3) | 0.679 |
| Pain-VAS | 12.3 (8.4–17.9) | 10.3 (5.4–14.8) | 0.012 |
| Fatigue score (0–100) | 64.2 (54.9–75.5) | 63 (51.4–74) | 0.287 |
| Patients with substantial morning stiffness[1] (above 15 minutes) | 2 (38) | 6 (5.2) | 1 |
| Charlson score | 1 (0.8–1) | 1 (0.8–1) | 0.614 |
| Patients on glucocorticoids[1] | 36 (67.9) | 62 (5.3) | 0.694 |
| Patients on DMARDs[1] | 53 (100) | 116 (100) | NA |
| Number of DMARDs/patient[2] | 2.8 (2.5–3.2) | 2.4 (2–2.8) | ≤0.0001 |
| Patients persistent in therapy[1] | 21 (39.6) | 39 (33.6) | 0.490 |

Data presents as median (p25-75) unless

[1] = N˚ (%) of patients.

[2] Among those with the characteristic.

*Nine and 13 missing data, respectively.

NA = not applicable.

predictors of incidental erosive disease and significant predictors when RF and ACPA were forced into the models. In model 1, baseline ESR, substantial morning stiffness, and ACPA predicted incidental erosive disease. In model 2, cumulative CRP and physician-VAS predicted incidental erosive disease. Finally, in model 3, ESR, ACPA, substantial morning stiffness (all at baseline), and cumulative CRP predicted incidental erosive disease.

## Discussion

The current study was performed in a well-characterized dynamic cohort of recent-onset RA patients, representative of real-life outpatients. All of them had prospective complete rheumatologic assessments that included physician evaluation, PROMs, biologic variables, and radiographic assessments on an annual basis. In addition, a directed evaluation at cohort entry of the real-life referral path to the cohort was also obtained, including time from symptoms onset, previous physicians' evaluation, and treatment. Data related to real-life referral paths are exceptionally retrieved but impact the clinical status of the patients at cohort enrollment. At the same time, it has been described that the level of disease activity at the baseline evaluation impacts future outcomes [27].

We first observed that 9% of the patients from the cohort were SNPs, and these patients showed a shorter time from symptoms onset to the first physician evaluation, visited fewer physicians, and received less intensive treatment in terms of glucocorticoids and DMARDs than SPPs, at the baseline evaluation and during the first two years of follow-up. Clinical outcomes and PROMs were similar at the baseline and during follow-up but for the cumulative

**Table 5. Multivariate Cox regression analysis to predict incidental erosive disease.**

| | Model 1 | Model 2 | Model 3 |
|---|---|---|---|
| | LLV = 521.799 | LLV = 489.600 | LVV = 483.390 |
| Baseline ESR | 1.019 (1.009–1.029), ≤0.0001 | | 1.013 (1.002–1.024), 0.021 |
| Baseline substantial morning stiffness at baseline | 10.961 (1.508–79.681), 0.018 | | 9.43 (1.291–68.875), 0.027 |
| Cumulative physician-VAS | | 1.057 (1.008–1.109), 0.023 | |
| Cumulative CRP | | 1.381 (1.012–1.885), 0.042 | 1.610 (1.224–2.118), 0.001 |
| *Baseline RF and ACPA forced into the models* | | | |
| | Model 1 | Model 2 | Model 3 |
| | LLV = 479.537 | | LVV = 472.069 |
| Baseline ESR | 1.017 (1.007–1.027), 0.001 | This is not applicable as only cumulative variables were considered in this model. | 1.011 (1.001–1.022), 0.034 |
| Baseline substantial morning stiffness at baseline | 11.296 (1.551–82.85), 0.017 | | 9.735 (1.33–71.243), 0.025 |
| Baseline positive ACPA | 10.654 (1.469–77.28), 0.019 | | 9.706 (1.336–70.516), 0.025 |
| Cumulative CRP | | | 1.544 (1.172–2.035), 0.002 |

Data are presented as HR, 95% CI, p. LLV = Log likelihood value.

Model 1: ESR (highly correlated to DAS28, rho = 0.733), CRP, substantial morning stiffness (>15 min), and the number of ACR 1987 classification criteria for RA, all at the baseline evaluation.

Model 2: Cumulative swollen joints (highly correlated to tender joints, rho = 0.870), ESR (highly correlated to DAS28, rho = 0.799), CRP, physician-VAS, and number of DMARDs/patient.

Model 3: baseline ESR, baseline substantial morning stiffness, cumulative CRP, and cumulative physician-VAS.

swollen joints (0–66), which were higher among SPPs who also referred to being more frequently persistent on therapy.

Our prevalence figure of seronegative RA is in the low range of that described in historical cohorts, 10–40% [28]. More recent studies, which include a collaboration of 16 registries, suggest a higher prevalence, up to 20–30% [4, 29], attributable to an aging population and its impact on RA epidemiology [4–6, 30]. Interestingly, a lower prevalence of seronegative RA has been observed in countries with low age at disease diagnosis [29], recognized as a differential characteristic of RA in LATAM populations [17]. However, data from early-onset cohorts highlight that up to 50–60% of the patients who fulfill RA classification criteria lack disease-specific autoantibodies [31, 32]. Differences observed in the estimated prevalence can be attributable to criteria selection cohorts and milder phenotypes described in some forms of seronegative RA [1]. We observed a lower percentage of SNPs currently active in the cohort. In comparison, a higher rate of patients was lost to follow-up, compared to SPPs, which might reflect self-remitting statuses in the former patients who might drop out of the cohort already described.

The natural history of seronegative RA is generally characterized by an abrupt onset, with a pattern of symptoms development that increases rapidly and is accompanied by muscle weakness [11, 12]. This might translate to the shorter time observed in our SNPs, from the symptom onset to the first physician evaluation, and the fewer physicians visited before cohort referral. Previous studies have confirmed longer, patient- and physician-related delays in SPPs, attributed to a more gradual symptom-onset and often "come-and-go" initial symptoms [11, 33–

35]. Also, in agreement with our result of higher cumulative swollen joints (0–66) in SPPs, there is published evidence that in ACPA-negative patients, symptoms start in the upper extremities [11]. In a large, prospective early arthritis cohort, no differences were observed in the first symptoms nor the signs found in the physical examination at initial presentation between ACPA-positive patients and negative patients; however, during follow-up, ACPA-positive patients had more swollen joints [36]. Finally, the differences observed between SPPs and SNPs in the treatment received and the treatment-related behavior can be explained by physicians' perception that seronegative RA is frequently milder or self-remitting and has lower DMARD requirements [1, 37, 38], more convincing benefits of some DMARDs in SPPs [1, 39], and lower retention rates of some DMARDs observed in SNPs [1, 9, 10].

Second, erosive disease was detected only in SPPs, and annual incidences increased during follow-up. At any time, erosions were detected more frequently in the feet, followed by the hands.

It has been well documented that SNPs achieve more frequent suppression of joint and systemic inflammation and remain erosive-free compared to SPPs [1, 38], confirmed in patients with recent-onset disease [15] and when bone erosions are detected with ultrasound instead of using conventional radiographs of hands and feet [40]. This association has been consistent in patients with ACPA seronegative status [36, 41–43]. At the same time, it has been observed that ACPA-positive patients develop erosions earlier and more abundantly than their counterparts [44]. Finally, similar to our results, previous studies had reported more erosions in the feet than in the hands among patients with recent-onset disease. Also, erosions developed earlier in the feet [45–47].

Baseline ACPA, ESR levels, substantial morning stiffness, and cumulative CRP were consistent predictors of incidental erosive disease in the different models tested.

Several studies have identified disease-specific autoantibodies as significant predictors of erosive disease [36, 43, 48, 49], which has also been confirmed among patients with recent-onset disease [40, 50, 51]. In their most recent report [51], Hetland et al. confirmed that positive ACPA (in addition to bone marrow edema) were independent predictors of radiographic progression after 11 years in 120 RA patients with early disease (< 6 months of symptoms duration). We did not identify RF as a significant predictor, similar to what has been published [52], and the differences observed might be due to the RA classification criteria used, time for erosive status assessment, how erosive status is assessed, and its characteristics (for instance, stable erosive status vs. progressive erosive status).

Baseline ESR has been identified as an independent predictor of radiographic progression. Syversen et al. [43] followed a cohort of 238 patients with RA longitudinally for ten years, among whom 125 patients had radiographic assessments of hands and feet at baseline and 10-year follow-up. ACPA, IgM RF, ESR, and female gender were independent predictors of radiographic progression and could be combined into an algorithm for better prediction.

(Significant) Morning joint stiffness has been proposed to be a marker of active disease that reflects the circadian pattern in the RA disease process, characterized by elevated nocturnal levels of pro-inflammatory cytokines that are insufficiently suppressed by endogenous nocturnal cortisol [53]. Interestingly, IL 6 seems instrumental in developing morning stiffness [53–55] and radiographic damage [56], which might explain our result. In addition, morning symptoms, including morning stiffness duration and severity, have been correlated with measures of disease activity such as DAS28, ACR20, and HAQ-DI, with the severity of morning stiffness showing greater effect size and less variability [57]. Ultimately, the level of disease activity at the onset is a significant negative prognostic factor [58, 59]. Our result has practical implications as morning stiffness-related information is a component of the medical history, easily obtained from the patient.

Finally, cumulative CRP predicted incidental erosive. Our findings support the established link between the prolonged effects of disease activity, as judged by CRP levels, and the progression of RA, as evidenced by radiologic assessments. This association has been observed in established [60–62] and early [63, 64] RA patients. A study by Van Leeuwen et al. [63] followed a cohort of 149 patients with symptoms duration of ≤1 year and found a correlation coefficient of 0.66 between radiologic progression and time-integrated CRP. Notably, CRP levels were more closely associated with radiologic progression than joint counts. The researchers also tried to account for individual variations in CRP levels by determining a mathematical constant that linked CRP levels to radiologic progression in each patient [64]. This allowed them to create a model that predicted joint damage at six years based on CRP and radiologic variables measured during the first six months of disease. It is important to note that CRP levels reflect an indirect measure of events in the synovium and that the total volume of synovial inflammation is what triggers the production of CRP.

Limitations of the study need to be addressed. First, this is a single-center study developed in a population of patients with particular characteristics, which limits results generalization. However, we consider it to contribute to current knowledge of the topic, which has been conceived based on studies performed mainly on Caucasians. Second, erosive disease status was established neither according to published definitions [52] nor using validated scoring methods but reflecting real-world physician evaluations; however, a radiologist and a rheumatologist read radiographs, and the increased prevalence of erosive disease observed during follow-ups, particularly in the feet location, is described when validated scoring systems are used. Third, non-erosiveness was established based on the absence of any erosion on conventional X-rays, and no additional techniques were used to confirm non-erosiveness status. Fourth, seronegative RA was defined based on the absence of serum RF and ACPA values above the local normal range; however, additional autoantibodies had been implicated in radiographic damage [3]. Fifth, the recent-onset cohort was initiated in 2004 when 2010 ACR/EULAR classification criteria were not available, and these perform better for recent-onset RA but still have insufficient sensitivity and specificity for seronegative forms [1]. Sixth, our definition of substantial morning stiffness was based on its duration over 15 minutes, and there is evidence that it might not be a marker of disease activity in early RA patients [65]. Seventh, the study was underpowered for baseline ESR to predict incidental erosive disease. Finally, a substantial number of patients were lost during follow-up, which was more evident in the seronegative group, which might have biased the results.

In conclusion, 9% of Mexican Mestizo patients from a recent onset dynamic RA cohort had concomitant negative RF and ACPA serum titers at baseline. These patients differed mildly in the real-life referral path, treatment received, and clinical outcomes from SPPs. However, only patients with disease-specific autoantibodies developed erosions over follow-up, which was more evident in the feet. In the cohort, the incidental erosive disease was predicted by baseline clinical (substantial morning stiffness), serologic (ESR and ACPA), and cumulative (CRP) variables.

## Supporting information

**S1 Checklist.** *PLOS ONE* **clinical studies checklist.**
(DOCX)

**S2 Checklist. STROBE checklist for cross sectional studies for paper "Differences in referral path, clinical and radiographic outcomes between seronegative and seropositive patients: A case control-study design within a dynamic recent-onset rheumatoid arthritis**

cohort".
(DOCX)

## Author Contributions

**Conceptualization:** Guillermo Arturo Guaracha-Basáñez, Irazú Contreras-Yáñez, Ana Belén Ortiz-Haro, Virginia Pascual-Ramos.

**Formal analysis:** Guillermo Arturo Guaracha-Basáñez, Irazú Contreras-Yáñez, Virginia Pascual-Ramos.

**Investigation:** Guillermo Arturo Guaracha-Basáñez, Irazú Contreras-Yáñez, Ana Belén Ortiz-Haro.

**Methodology:** Guillermo Arturo Guaracha-Basáñez, Irazú Contreras-Yáñez, Virginia Pascual-Ramos.

**Project administration:** Virginia Pascual-Ramos.

**Software:** Irazú Contreras-Yáñez.

**Supervision:** Irazú Contreras-Yáñez, Virginia Pascual-Ramos.

**Validation:** Guillermo Arturo Guaracha-Basáñez, Irazú Contreras-Yáñez, Ana Belén Ortiz-Haro, Virginia Pascual-Ramos.

**Visualization:** Guillermo Arturo Guaracha-Basáñez, Virginia Pascual-Ramos.

**Writing – original draft:** Virginia Pascual-Ramos.

**Writing – review & editing:** Guillermo Arturo Guaracha-Basáñez, Irazú Contreras-Yáñez, Ana Belén Ortiz-Haro, Virginia Pascual-Ramos.

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
