## [Decision Letter · Decision Letter 0]

8 Apr 2024

PONE-D-23-43711Differences in referral path, clinical and radiographic outcomes between seronegative and seropositive patients: A case control-study design within a dynamic recent-onset rheumatoid arthritis cohort.PLOS ONE

Dear Dr. Pascual-Ramos,

Thank you for submitting your manuscript to PLOS ONE. After careful consideration, we feel that it has merit but does not fully meet PLOS ONE’s publication criteria as it currently stands. Therefore, we invite you to submit a revised version of the manuscript that addresses the points raised during the review process.

We look forward to receiving your revised manuscript.

Kind regards,

Jan René Nkeck, M.D., M.Sc

Academic Editor

PLOS ONE

3. In the online submission form, you indicated that [The data that support the findings of this study are not openly available due to reasons of sensitivity and privacy, however are available from the corresponding author upon reasonable request and with approval of the local IRB.]. 

Reviewers' comments:

Reviewer's Responses to Questions

**Comments to the Author**

1. Is the manuscript technically sound, and do the data support the conclusions?

Reviewer #1: Partly

Reviewer #2: Partly

2. Has the statistical analysis been performed appropriately and rigorously? 

Reviewer #1: No

Reviewer #2: Yes

3. Have the authors made all data underlying the findings in their manuscript fully available?

Reviewer #1: Yes

Reviewer #2: No

4. Is the manuscript presented in an intelligible fashion and written in standard English?

Reviewer #1: Yes

Reviewer #2: Yes

5. Review Comments to the Author

Reviewer #1: I have had the opportunity to review the manuscript entitled: “Differences in referral path, clinical and radiographic outcomes between seronegative and seropositive patients: A case control-study design within a dynamic recent-onset rheumatoid arthritis cohort”, whose fundamental purpose is to compare clinical and sociodemographic features, besides radiological damage, among patients with rheumatoid arthritis seropositive to RF and/or ACPA against those negative to both auto-antibodies.

The study addresses an extremely relevant topic for the care of patients with RA and I consider that it may be suitable for publication in your journal. However, from my point of view the masuscript must be subjected to a series of corrections and clarifications, both exhaustive, since its current version has relevant aspects of form and substance that make it be at high risk of having systematic deviations that significantly detract from its content.

The first aspect: I do not see the reason, or the justification, for having downgraded the explanatory level of the methodological design to a case-control study (even if it is nested), since the authors have an adequate and well-run inception cohort capable of giving the required information with greater explanatory power and strength of conclusions than a case-control study. Additionally, it is well established that cohort analysis (even retrospective or “trohoc”) is the best design for conducting studies with the purpose of delineating the clinical course, prognosis, and prognostic factors such as the current one. Derived from all this, my suggestion is that the study be focused and analyzed as a cohort study, instead of cases and controls, given that the authors have all the resources to carry out this conversion. In this case the exposure would be the baseline seropositivity, the outcome the erosive disease and the intervening factors the other data collected.

Below I present specific aspects of the manuscript:

Title: Given that the study has a case-control design, defining the case as “erosive disease”, the title should prioritize this fact, instead of prioritizing the seropositivity status.

Abstract:

The authors take the terms of radiographic damage and erosive disease as synonyms, when in the strict sense they are not. There are other findings of radiographic damage (for example, decreased joint space and luxations) besides erosions.

The usual numerical indicator of statistical association for logistic regression analyzes is the odds ratio; the hazard ratio corresponds to statistical analyzes that adjust for unequal follow-up times, such as the Cox proportional hazards model.

There is a contradiction in the presentation of the results, on the one hand, it is mentioned that the erosive disease was more frequently associated with seropositivity at two and five years of follow-up, and in the immediately following paragraph it is stated that the autoantibodies did not predict the radiographic outcomes . The conclusions insist on this fact.

Introduction:

Authors generally present the state of the art regarding the knowledge of the differences between seropositivity and seronegativity in RA and establish that seropositivity has already been widely identified as a predictor of radiographic damage. This detracts from the originality of the central reason for the study (according to the authors), therefore, I consider that in this section we should delve more into the aspect of originality as a criterion for scientific justification of the study.

Material and methods:

The description of the participants reveals that it is indeed a successful inception cohort, with a systematic collection, both initially and throughout follow-up, of all the data necessary for an adequate characterization of the relevant characteristics of the condition.

In the study design and data collection section: It begins by commenting that “the present study has a cohort design” in the title and mentions that it is a case-control study. The authors should clarify this fact. The clarification of this point is not trivial, since it directly affects the statistical analysis to be carried out.

In the definitions section: Were the readers of the radiographic studies blinded to the clinical context of the patients?

Results:

Although they constitute a significant extension of text and data (which makes them difficult to read), they are all important, given the characteristics and objectives of the study. In general, all results are presented correctly, and are referred to appropriately for tabular query.

Discussion:

It is adequate in extent; discusses, compares, and tries to adequately explain the fundamental findings of the study with the literature reports relevant to each of them.

A fundamental aspect that the authors omit in the discussion and that I consider an important limitation of the study, and that must be incorporated into the discussion: The lack of significance of exposure to seropositivity at baseline as a prognostic factor for erosive disease at 2 and 5 years of follow-up. This may be due to two factors:

The most important is the high loss of subjects throughout the study; The total number of participants in the cohort is 237, of which 188 (79.3%) were analyzed at 2 years, and 144 (60.7%) at 5 years of follow-up. Additionally, the authors do not mention in the results the proportion of seronegative patients in the total cohort (237 patients), so that the reader can have a clear idea, taking into account the subjects available for analysis at 2 and 5 years, if there was a differential rate of loss between seropositive and seronegative patients at these two cut-off points. Taking all of the above into account, it is likely that the loss of significance of seropositivity status as a prognostic factor for erosive disease is due to the high rate of losses in the total cohort, and possibly, more marked in the seronegative subcohort, it may have been resulted in the “survivor cohort” bias.

The other aspect that can explain the fact already mentioned is the type of analysis carried out. Instead of having performed a logistic regression model (which does not adjust for differential follow-up times), the authors could have performed an analysis using the Cox proportional hazards method, which does use the harzard ratio as a numerical indicator of association and adjusts for censored patients and with differential follow-up time. This could have resulted in the possibility of entering a larger sample size and giving greater statistical power to the study, since it would give the possibility of including patients with a shorter follow-up time, which would give the possibility of including patients who developed erosions during the first year of follow-up.

Reviewer #2: Representing data regarding seropositive and seronegative RA in a Mexican cohort is interesting.

Addressing the differences in referral path clinical, laboratory and radiographic characteristics seems a valid hypotheses, however, there are important comments that require to be addressed in the methodology.

- The methodology for the sample size calculation is missing and if not done the authors need to clarify the reason.

- It is of utmost importance to discuss in details how was radiographic assessment done using which radioimaging technique.

- It is well identified in RA that assessing erosion in RA requires detailed explanation of the scores measures and their numerical values of course in addition to the radioimaging technique used to count erosion?

- Exploring the relationship between seropositivity and erosion requires all what was mentioned above to provide valid data?

- It is clear that seropositivity seems related to aggressive disease in the studied cohort?

- Despite the variation in disease activity status the details regarding the types of DMARDs, their combination and their dosages requires further emphasis.

- The reported data are very interesting yet conclusions are liable to be affected by a number of confounded and limitations, how were they handled this wasn't discussed in the discussion.

6. PLOS authors have the option to publish the peer review history of their article (what does this mean?). If published, this will include your full peer review and any attached files.

Reviewer #1: No

Reviewer #2: No

---

## [Author Response · Author response to Decision Letter 0]

24 Apr 2024

Responses to reviewers

Reviewer #1: I have had the opportunity to review the manuscript entitled: “Differences in referral path, clinical and radiographic outcomes between seronegative and seropositive patients: A case control-study design within a dynamic recent-onset rheumatoid arthritis cohort”, whose fundamental purpose is to compare clinical and sociodemographic features, besides radiological damage, among patients with rheumatoid arthritis seropositive to RF and/or ACPA against those negative to both auto-antibodies.

The study addresses an extremely relevant topic for the care of patients with RA and I consider that it may be suitable for publication in your journal. However, from my point of view the manuscript must be subjected to a series of corrections and clarifications, both exhaustive, since its current version has relevant aspects of form and substance that make it be at high risk of having systematic deviations that significantly detract from its content.

The first aspect: I do not see the reason, or the justification, for having downgraded the explanatory level of the methodological design to a case-control study (even if it is nested), since the authors have an adequate and well-run inception cohort capable of giving the required information with greater explanatory power and strength of conclusions than a case-control study. Additionally, it is well established that cohort analysis (even retrospective or “trohoc”) is the best design for conducting studies with the purpose of delineating the clinical course, prognosis, and prognostic factors such as the current one. Derived from all this, my suggestion is that the study be focused and analyzed as a cohort study, instead of cases and controls, given that the authors have all the resources to carry out this conversion. In this case the exposure would be the baseline seropositivity, the outcome the erosive disease and the intervening factors the other data collected.

Response: We agree with the reviewer and have adopted the suggestion. The manuscript's updated version proposes a Cox regression analysis to define HR and 95% CI for RF and ACPA (and additional factors) to predict INCIDENTAL erosive disease. Also, we acknowledge in the study limitations that bias derived from losses to follow-up is inherent to a cohort design. 

Below I present specific aspects of the manuscript:

Title: Given that the study has a case-control design, defining the case as “erosive disease”, the title should prioritize this fact, instead of prioritizing the seropositivity status.

Response: In the updated version of the manuscript, we are no longer proposing a case-control design and have accordingly updated the title to “Differences in referral path, clinical and radiographic outcomes between seronegative and seropositive rheumatoid arthritis Hispanic patients: A cohort study.” 

Abstract:

The authors take the terms of radiographic damage and erosive disease as synonyms, when in the strict sense they are not. There are other findings of radiographic damage (for example, decreased joint space and luxations) besides erosions.

Response: We agree with the reviewer and have been more precise regarding the radiographic outcome observed: erosive disease and incidental ersove disease. 

The usual numerical indicator of statistical association for logistic regression analyzes is the odds ratio; the hazard ratio corresponds to statistical analyzes that adjust for unequal follow-up times, such as the Cox proportional hazards model.

Response: We agree with the reviewer. In the manuscript's updated version, we propose Cox proportional hazards models. 

There is a contradiction in the presentation of the results, on the one hand, it is mentioned that the erosive disease was more frequently associated with seropositivity at two and five years of follow-up, and in the immediately following paragraph it is stated that the autoantibodies did not predict the radiographic outcomes. The conclusions insist on this fact.

Response: The current version includes the reviewer's suggestions (study design, multivariate Cox regression analysis) and evidence that ACPA might predict INCIDENTAL erosive disease. The abstract has been updated and offers no contradictions (hopefully). 

Introduction:

Authors generally present the state of the art regarding the knowledge of the differences between seropositivity and seronegativity in RA and establish that seropositivity has already been widely identified as a predictor of radiographic damage. This detracts from the originality of the central reason for the study (according to the authors), therefore, I consider that in this section we should delve more into the aspect of originality as a criterion for scientific justification of the study.

Response: We have updated the introduction section to highlight that most of the current seronegative RA knowledge is being conceived based on studies performed on Caucasians. Meanwhile, it is currently evident that (primarily seropositive) RA in patients from Latin American countries displays differential characteristics that are not limited to socio-demographics. These differential phenotypes might also extend to “seronegative” RA. Accordingly, there is a gap in seronegative RA knowledge and studies, and the current study helps to narrow this gap. 

Material and methods:

The description of the participants reveals that it is indeed a successful inception cohort, with a systematic collection, both initially and throughout follow-up, of all the data necessary for an adequate characterization of the relevant characteristics of the condition.

Response: Thank you for the comment. 

In the study design and data collection section: It begins by commenting that “the present study has a cohort design” in the title and mentions that it is a case-control study. The authors should clarify this fact. The clarification of this point is not trivial, since it directly affects the statistical analysis to be carried out.

Response: We have updated the whole manuscript. The study is a cohort study. All the sections have been updated to conform with a cohort design and a multivariate Cox regression analysis is proposed to identify predictors of incidental erosive disease. 

In the definitions section: Were the readers of the radiographic studies blinded to the clinical context of the patients?

Response: We have clarified in the definition section that physicians were not blinded to the patient’s clinical context. 

Results:

Although they constitute a significant extension of text and data (which makes them difficult to read), they are all important, given the characteristics and objectives of the study. In general, all results are presented correctly, and are referred to appropriately for tabular query.

Response: Thank you for the comment. 

Discussion:

It is adequate in extent; discusses, compares, and tries to adequately explain the fundamental findings of the study with the literature reports relevant to each of them.

Response: Thank you for the comment. 

A fundamental aspect that the authors omit in the discussion and that I consider an important limitation of the study, and that must be incorporated into the discussion: The lack of significance of exposure to seropositivity at baseline as a prognostic factor for erosive disease at 2 and 5 years of follow-up. This may be due to two factors:

The most important is the high loss of subjects throughout the study; The total number of participants in the cohort is 237, of which 188 (79.3%) were analyzed at 2 years, and 144 (60.7%) at 5 years of follow-up. Additionally, the authors do not mention in the results the proportion of seronegative patients in the total cohort (237 patients), so that the reader can have a clear idea, taking into account the subjects available for analysis at 2 and 5 years, if there was a differential rate of loss between seropositive and seronegative patients at these two cut-off points. Taking all of the above into account, it is likely that the loss of significance of seropositivity status as a prognostic factor for erosive disease is due to the high rate of losses in the total cohort, and possibly, more marked in the seronegative subcohort, it may have been resulted in the “survivor cohort” bias.

The other aspect that can explain the fact already mentioned is the type of analysis carried out. Instead of having performed a logistic regression model (which does not adjust for differential follow-up times), the authors could have performed an analysis using the Cox proportional hazards method, which does use the hazard ratio as a numerical indicator of association and adjusts for censored patients and with differential follow-up time. This could have resulted in the possibility of entering a larger sample size and giving greater statistical power to the study, since it would give the possibility of including patients with a shorter follow-up time, which would give the possibility of including patients who developed erosions during the first year of follow-up.

Response: We highly appreciate the reviewer's comments and criticisms. We have updated the study design and performed a Cox regression analysis as suggested. The discussion has been updated according to the results obtained. The current version of the manuscript is much better, and we sincerely acknowledge the reviewer's commitment to improving the manuscript. 

Reviewer #2: Representing data regarding seropositive and seronegative RA in a Mexican cohort is interesting.

Addressing the differences in referral path clinical, laboratory and radiographic characteristics seems a valid hypothesis, however, there are important comments that require to be addressed in the methodology.

- The methodology for the sample size calculation is missing and if not done the authors need to clarify the reason.

Response: We have added a paragraph suitable for a cohort study design. 

- It is of utmost importance to discuss in details how was radiographic assessment done using which radioimaging technique.

- It is well identified in RA that assessing erosion in RA requires detailed explanation of the scores measures and their numerical values of course in addition to the radioimaging technique used to count erosion?

- Exploring the relationship between seropositivity and erosion requires all what was mentioned above to provide valid data

Response: We have updated the corresponding section with the information required. Also, we have addressed as a limitation of the study that no validated scoring system was used to define erosive disease. 

- It is clear that seropositivity seems related to aggressive disease in the studied cohort?

Response: We agree with the reviewer; it is particularly true regarding the erosive disease. Only mild differences were observed in the referral path, clinical outcomes (0-66 swollen joints), and treatment. The current version of the manuscript proposes a cohort study design and uses multivariate Cox regression analysis (suggested by one reviewer) to evidence predictors of incidental erosive disease. Increased erosive disease prevalence at follow-ups is evident among seropositive patients, and ACPA positivity appears as a predictor of incidental erosive disease.

- Despite the variation in disease activity status the details regarding the types of DMARDs, their combination and their dosages requires further emphasis.

Response: We have updated the results section with some information regarding the most frequent DMARD and DMARD combinations used. We apologize, but we do not have data on DMARD doses.

- The reported data are very interesting, yet conclusions are liable to be affected by a number of confounded and limitations; how were they handled? This wasn't discussed in the discussion.

Response: We have updated the discussion, limitations, and conclusions sections according to the new statistical analysis performed. The latest proposal better addresses limitations and how we handled them.

---

## [Decision Letter · Decision Letter 1]

16 May 2024

PONE-D-23-43711R1Differences in referral path, clinical and radiographic outcomes between seronegative and seropositive rheumatoid arthritis Hispanic patients: A cohort study.PLOS ONE

Dear Dr. Pascual-Ramos,

Thank you for submitting your manuscript to PLOS ONE. After careful consideration, we feel that it has merit but there are still a few minor suggestions we'd like you to address before publication. Therefore, we invite you to submit a revised version of the manuscript that addresses the points raised during the review process.

We look forward to receiving your revised manuscript.

Kind regards,

Jan René Nkeck, M.D., M.Sc

Academic Editor

PLOS ONE

Journal Requirements:

Reviewers' comments:

Reviewer's Responses to Questions

**Comments to the Author**

1. If the authors have adequately addressed your comments raised in a previous round of review and you feel that this manuscript is now acceptable for publication, you may indicate that here to bypass the “Comments to the Author” section, enter your conflict of interest statement in the “Confidential to Editor” section, and submit your "Accept" recommendation.

Reviewer #1: All comments have been addressed

2. Is the manuscript technically sound, and do the data support the conclusions?

Reviewer #1: Yes

3. Has the statistical analysis been performed appropriately and rigorously? 

Reviewer #1: Yes

4. Have the authors made all data underlying the findings in their manuscript fully available?

Reviewer #1: Yes

5. Is the manuscript presented in an intelligible fashion and written in standard English?

Reviewer #1: Yes

6. Review Comments to the Author

Reviewer #1: Dear Editor:

I have reviewed the corrected version of the manuscript now titled “Differences in referral path, clinical and radiographic outcomes between seronegative and seropositive rheumatoid arthritis Hispanic patients: A cohort study.” and I consider that the authors have carried out, from my point of view, the suggested changes, or have considered not making changes based on well-founded arguments, so I consider that the manuscript could be eligible for acceptance by your journal. However, the conditionality of acceptance would be subject to making some minor changes that, from my point of view, would increase the scientific quality of the content of the manuscript.

Since these changes are minor, I consider that the manuscript could be accepted without further revision on my part, if you consider that these minimal changes were carried out by the authors.

These changes are:

Consider whether the ethnic background of the target population is more accurately described under the term “Mexican Mestizo,” rather than “Hispanic,” both in the title and throughout the content.

There are small discrepancies in the percentage frequencies of patients who developed erosive disease in the summary, in the text of the results, in the corresponding table, and in the discussion. These percentage frequencies must be unified.

Correct the term of the epidemiological indicator for the percentage frequencies of erosive disease. At baseline, the term “prevalence” is appropriate, but from the first year onwards, given that it is the result of outcomes that occur throughout the follow-up, the appropriate term, from my point of view, is incidence.

In the introduction (page 14 of manuscript, line 88), I consider that the most appropriate term is “Caucasian” instead of “White population”.

Consider changing the location of Table 1 from the Materials and Methods section to the Results section, since these are actually baseline results.

In the first line of the “Radiographic outcomes” section (Table 3) of the Results, I consider that it should be specified that the erosive disease was detected only in seropositive patients, both at baseline and throughout follow-up. Additionally, the final phrase of this section that says, “Also, more SPPs had erosions compared to SNPs, and these differences reached statistical significance since the first years of follow-up and at the feet location .” may be repetitive, since from the first line of this paragraph it was specified that erosions were only detected in seropositive patients.

In the heading of Table 4, I consider that the adjective “incidental” should be added to the term “erosive disease”, since this outcome is what this table specifically describes.

7. PLOS authors have the option to publish the peer review history of their article (what does this mean?). If published, this will include your full peer review and any attached files.

Reviewer #1: **Yes: **Jose Alvarez-Nemegyei

---

## [Author Response · Author response to Decision Letter 1]

20 May 2024

Reviewer #1: Dear Editor:

I have reviewed the corrected version of the manuscript now titled “Differences in referral path, clinical and radiographic outcomes between seronegative and seropositive rheumatoid arthritis Hispanic patients: A cohort study.” and I consider that the authors have carried out, from my point of view, the suggested changes, or have considered not making changes based on well-founded arguments, so I consider that the manuscript could be eligible for acceptance by your journal. However, the conditionality of acceptance would be subject to making some minor changes that, from my point of view, would increase the scientific quality of the content of the manuscript.

Since these changes are minor, I consider that the manuscript could be accepted without further revision on my part, if you consider that these minimal changes were carried out by the authors.

These changes are:

Consider whether the ethnic background of the target population is more accurately described under the term “Mexican Mestizo,” rather than “Hispanic,” both in the title and throughout the content.

Response. We have adopted the suggestion.

There are small discrepancies in the percentage frequencies of patients who developed erosive disease in the summary, in the text of the results, in the corresponding table, and in the discussion. These percentage frequencies must be unified.

Response. We apologize for the mistake and have unified the numbers along the text. 

Correct the term of the epidemiological indicator for the percentage frequencies of erosive disease. At baseline, the term “prevalence” is appropriate, but from the first year onwards, given that it is the result of outcomes that occur throughout the follow-up, the appropriate term, from my point of view, is incidence.

Response. We have adopted the suggestion. 

In the introduction (page 14 of manuscript, line 88), I consider that the most appropriate term is “Caucasian” instead of “White population”.

Response. We have adopted the suggestion. 

Consider changing the location of Table 1 from the Materials and Methods section to the Results section, since these are actually baseline results.

Response. We have adopted the suggestion.

In the first line of the “Radiographic outcomes” section (Table 3) of the Results, I consider that it should be specified that the erosive disease was detected only in seropositive patients, both at baseline and throughout follow-up. Additionally, the final phrase of this section that says, “Also, more SPPs had erosions compared to SNPs, and these differences reached statistical significance since the first years of follow-up and at the feet location.” may be repetitive, since from the first line of this paragraph it was specified that erosions were only detected in seropositive patients.

Response. We have updated the paragraph to adopt the suggestions. 

In the heading of Table 4, I consider that the adjective “incidental” should be added to the term “erosive disease”, since this outcome is what this table specifically describes.

Response. We have adopted the suggestion.

---

## [Editor Report · Decision Letter 2]

22 May 2024

Differences in referral path, clinical and radiographic outcomes between seronegative and seropositive rheumatoid arthritis Mexican Mestizo patients: A cohort study.

PONE-D-23-43711R2

Dear Dr. Pascual-Ramos,

We’re pleased to inform you that your manuscript has been judged scientifically suitable for publication and will be formally accepted for publication once it meets all outstanding technical requirements.

Kind regards,

Jan René Nkeck, M.D., M.Sc

Academic Editor

PLOS ONE

---

## [Editor Report · Acceptance letter]

27 May 2024

PONE-D-23-43711R2 

PLOS ONE

Dear Dr. Pascual-Ramos, 

I'm pleased to inform you that your manuscript has been deemed suitable for publication in PLOS ONE. Congratulations! Your manuscript is now being handed over to our production team.

Kind regards, 

on behalf of

Dr. Jan René Nkeck 

Academic Editor

PLOS ONE